# Improving the "real life" management of schizophrenia spectrum disorders by LAI antipsychotics: A one-year mirror-image retrospective study in community mental health services

**Valeria Latorre**[1]*, **Apostolos Papazacharias**[1], **Maria Lorusso**[1], **Gaetano Nappi**[1], **Paola Clemente**[1], **Antonia Spinelli**[1], **Giovanni Carrieri**[1], **Enrico D'Ambrosio**[2], **Michele Gattullo**[3], **Antonio Emmanuele Uva**[3], **Domenico Semisa**[1]

1 Department of Mental Health, Azienda Sanitaria Locale Bari, Bari, Italy, 2 Department of Basic Medical Science, Neuroscience and Sense Organs, University of Bari Aldo Moro, Bari, Italy, 3 Department of Mechanics, Mathematics and Management, Polytechnic Institute of Bari, Bari, Italy

* valeria.latorre@asl.bari.it

## Abstract

Schizophrenia poses a significant economic burden on the healthcare system as well as it has a significant impact on society at large. Reasons for such a high economic burden of schizophrenia include the frequent relapses and hospitalizations occurring in this disorder. We analyze the effectiveness of long-acting injectable antipsychotics (LAIs) compared to oral medications, in terms of "clinical process management" in a sample of patients with a diagnosis of schizophrenia spectrum disorder treated in community mental health centers. An observational, retrospective, mirror-image study was carried out to evaluate the effectiveness of LAIs compared to oral medications in terms of number of hospitalizations, emergency visits and planned visits on a 10-year period (from July 2007 to June 2017). Differences between first and second generation LAIs were also explored. Our findings show that hospitalization and emergency visits are significantly decreased with the use of LAIs, while planned visits are increased in patients treated with LAIs. Our results suggest that LAIs, in particular, second generation ones, reduce hospitalization rates and emergency visits, improving the economic burden of schizophrenia. Therefore, LAIs should be considered a cost-effective treatment in the management of schizophrenia under routine conditions.

## Introduction

Schizophrenia is a severe, chronic, often recurring mental disorder affecting 1% of the general population, and it is associated with a relevant long-term impact on patients' social and occupational functioning. It is treated with a combination of medical, psychological and psychosocial interventions, with varying degrees of success. The economic consequences of

**Data Availability Statement:** All relevant data are within the manuscript and its Supporting Information files.

**Funding:** The authors received no specific funding for this work.

**Competing interests:** The authors have declared that no competing interests exist.

schizophrenia, defined as costs of illness generated by the aggregation of direct and indirect costs [1], are considerable. Direct costs include those associated with inpatient (i.e., hospitalizations) and outpatient treatments, long-term care, costs of medications, and justice costs. Indirect costs arising from loss of productivity suffered by individuals with schizophrenia and their family members. The main reasons for such a high economic burden of this disorder are complex clinical processes related to the early onset, the chronic nature with frequent relapses and high rates of hospitalizations [2]. Complex clinical processes include all activities provided by healthcare professionals addressing patients' healthcare issues, that refer not only to hospitalizations but also to emergency and planned outpatient visits. In order to increase the quality of care and to reduce treatment costs, it is of paramount importance to optimize those clinical processes [3].

Non-adherence to antipsychotic medications is one of the most important factors increasing relapses in schizophrenia [4,5]. About 60% of patients with schizophrenia are non-adherent to antipsychotic medications already in the first phases of the illness and are less likely to be compliant later on [2]. Most guidelines for the management of schizophrenia recommend improving medication adherence as a strategy to reduce hospitalization rates and costs [6]. A systematic review and meta-analysis of 25 mirror-image studies in patients eligible for clinical use of LAIs showed strong superiority of LAIs compared to oral antipsychotics in preventing hospitalization [7]. These results are in contrast to the meta-analysis of randomized controlled trials (RCTs), which showed no superiority of LAIs in preventing relapse and hospitalizations [8].

Another recent study, carried on an administrative database analysis in Japan [9], showed that LAIs, compared to oral antipsychotics, reduce rehospitalizations and emergency visits. A very recent Swedish study [10] analyzed 29,823 patients with schizophrenia from nation-wide register-based data to evaluate the risk of rehospitalization and treatment failure. The authors found that clozapine and LAIs are the pharmacological treatments with the highest rate of relapse prevention. However, most of the studies have been conducted under controlled conditions and have evaluated rehospitalization rates only.

Given the possible biases in mirror-image studies, such as expectation biases, natural illness course, and time-effect, a cautious interpretation is required. Nevertheless, the population in mirror–image studies better reflects the population receiving LAIs in clinical practice [7]. In this study, our goal is not to test the efficacy of LAIs compared to oral medications, but we aim to evaluate the effectiveness (i.e., efficacy under ordinary circumstances) in terms of clinical process management in specific patients, with a diagnosis of schizophrenia spectrum disorder, who needed to switch from oral to LAI therapy in real-life conditions [7]. In order to obtain real-life measures, patients had to be treated in community mental health centers. The effectiveness of antipsychotic medications was evaluated through means of hospitalizations, emergency and planned visits.

## Material and methods

### Study design

An observational, retrospective, naturalistic, mirror-image study was designed to determine the efficacy of LAIs compared to oral antipsychotics. The use of mirror-image study design does not include a parallel active control group; instead, each patient serves as their own control.

As a result, it cannot be determined whether other treatments may have had similar effects. We defined Time 0 (T0) as the time in which each patient switched from oral to LAI antipsychotic medication. Patients were recruited in 5 community mental health services of the

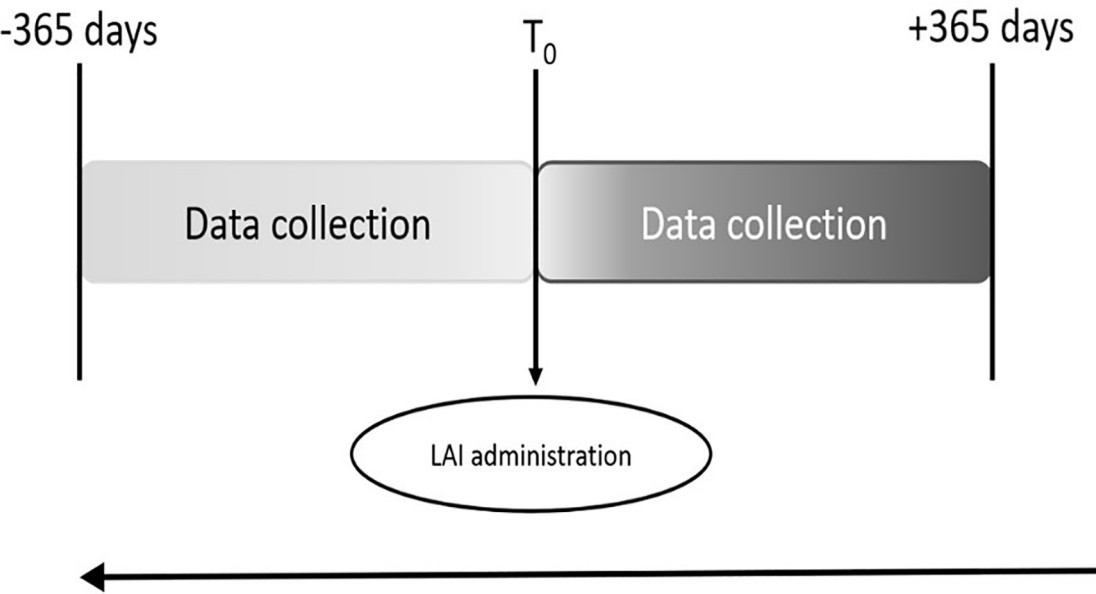

**Fig 1. Study design.**

Department of Mental Health of Bari. We informed the local ethical committee prior to initiating the study, in line with Istituto Superiore di Sanità protocol. Each patient was assigned with an ID code to guarantee anonymity. All the patients whose data were collected had previously signed the informed consent, present in the medical record, to the processing of personal data and the use of the data for research purposes. Given the naturalistic design of the study, the results remained purely observational and researchers did not influence the results in any way. The study design is detailed in Fig 1.

## Study sample

The clinical and electronic (SISM Experia, Italy) files of all patients attending five community mental health services of the Department of Mental Health of Bari (ASL BA) and receiving LAI antipsychotic medications from July 2007 to June 2017 were analyzed.

Exclusion criteria were: a) diagnosis of schizophrenia spectrum disorder according to the DSM-5 criteria for less than one year before T0; b) LAI concomitant antipsychotic medication; c) substance use disorder or of intellectual disability disorder; d) a major change in life situation (i.e., admission in residential facilities programs) in the year before and after T0. All patients had been treated with oral antipsychotics one year before T0 and with LAIs for one year after T0. Patients with illegible medical records were excluded.

## Study measures and end-points

Patients' demographic characteristics, including age, gender, educational level, diagnosis and sub-diagnosis of schizophrenia spectrum disorder, duration of illness at T0 and oral psycho-pharmacological medications before and after T0, were registered for all patients included in the analyses. For all patients, the following information was collected one year before and one year after T0: type of LAI antipsychotic treatment (in particular first or second generation antipsychotic); number of hospitalizations; number of emergency visits; number of planned visits. We defined the primary end-points of the study: (i) hospitalization rates, (ii) total number of

hospitalizations, (iii) emergency rates, and (iv) total number of emergency visits, as associated with the severity of the condition of the patient. As secondary end-point, we considered the (v) total number of planned outpatient visits, as associated with the therapeutic compliance and alliance of the patient. In the number of planned outpatient visits, we excluded the planned contacts with nurses for injection administration. Hospitalization rates were calculated as the proportion of patients with ≥ one psychiatric hospitalization [1]. Similarly, emergency rates were calculated as the proportion of patients with ≥ one emergency visits. First and second-generation LAI antipsychotics were compared one year before and one year after T0 on all assessed outcome measures.

### Statistical analyses

Effects of treatment (before and after T0), LAI generation, age, gender, educational level, diagnosis, and illness duration on the end-points were measured. We used GEE (Generalized Estimating Equations) models to account for within-subject correlations. These preliminary analyses revealed that the treatment and LAI generation seems to have major effects. Therefore, we studied in deep with specific tests the effects of treatment and LAI generation.

Non-parametric tests were used since the distribution of the dependent variables (hospitalization rates, total number of hospitalizations, total number of planned outpatient visits, emergency rates, and total number of emergency visits) was non-normal. The McNemar test was used to study the effect of LAIs on hospitalization and emergency rates in order to determine if the number of patients that were hospitalized / required emercency visits before T0 (dependent variables: "hospitalization"/ "emergency"; "yes" or "no" categories) decreased after the introduction of LAIs. Wilcoxon test was used to study the effect of LAIs on total number of hospitalizations, total number of planned outpatient visits, and total number of emergency visits before and after T0. The Mann-Whitney U test was used to compare the differences between first and second-generation antipsychotics on the four analyzed dependent variables.

To establish the real-world benefit, we plan to test the hypothesis that it is necessary to demonstrate the LAI effects on "all" the primary endpoints. If this hypothesis is rejected, we can test the weaker hypothesis that the demonstration of the LAI effect on at least one of several primary endpoints is sufficient. In this case, correction for multiple comparisons should be performed to control the Type I error. We chose to apply the Holm–Bonferroni method, if the case.

## Results

### Subjects

Data from 207 patient records were collected (Table 1). Sixty percent of them were male, with a mean age of 47.9 (SD 12.0) years, a mean duration of illness of 15.8 (SD 8.7) years, a mean educational level of 9.2 (SD 3.6) years. They had a diagnosis of schizophrenia (48%), schizoaffective disorder (30%), or other specified schizophrenia spectrum and other psychotic disorders (22%). 49% of patients were in monotherapy with LAIs (Fig 2). At T0, 68% of patients were treated by second generation LAIs (paliperidone: 27.5%; risperidone: 22.2%; aripiprazole: 13.5%; olanzapine: 4.8%) and 32% by first generation (haloperidol: 14.0%; fluphenazine: 11.6%; zuclopenthixol: 4.8%; perphenazine: 1.4%).

### GEE models

GEE model analyses, whose results are reported in Table 2, revealed that the treatment has an effect on all the end-points. LAI generation seems to have an effect on hospitalization rates, emergency rates, and total number of emergency visits; it has a lower effect on total number of

**Table 1. Sociodemographic and clinical characteristics of our sample.**

| Characteristics | LAI 1 (n = 66) | LAI 2 (n = 141) |
|---|---|---|
| **Gender, n (%)** | | |
| Male | 45 (68.2%) | 79 (56.0%) |
| Female | 21 (31.8%) | 62 (44.0%) |
| **Age, mean ±SD** | | |
| Years | 54.1 ± 10.8 | 44.9 ± 11.5 |
| **Educational level, mean ±SD** | | |
| Years of scholarization | 8.4 ± 3.4 | 9.5 ± 3.7 |
| **Diagnosis, n (%)** | | |
| Schizophrenia | 36 (54.5%) | 64 (45.4%) |
| Schizoaffective disorder | 17 (25.8%) | 45 (31.9%) |
| Other schizophrenia spectrum | 13 (19.7%) | 32 (22.7%) |
| **Illness duration, mean ±SD** | | |
| Years | 20.5 ± 7.4 | 13.6 ± 8.4 |

hospitalizations and no effect on total number of planned outpatient visits. As to the other characteristics, there is not enough evidence to conclude that they have an effect on the end-points.

## LAI versus oral antipsychotic treatment

After switching to LAIs, the number of hospitalizations was drastically reduced from 187 (0.90 hospitalizations per patient/year) to 20 (0.10 hospitalizations per patient/year) (Wilcoxon test;

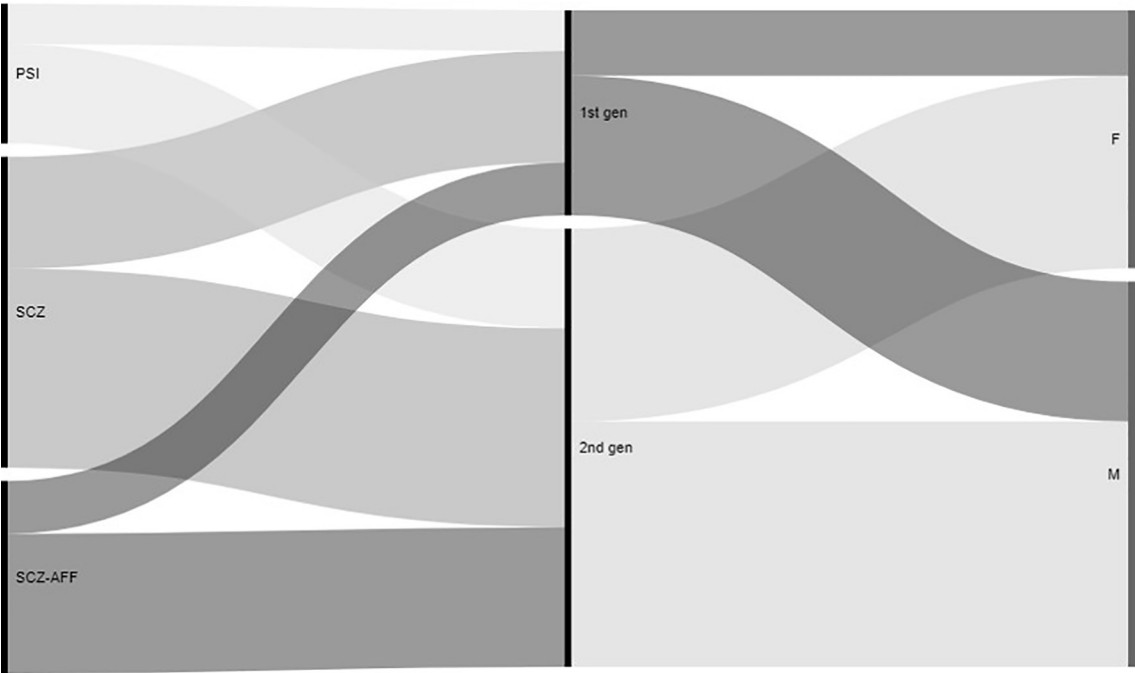

**Fig 2. Diagnosis: Schizophrenia (SCZ) 48%, Schizoaffective disorder (SCZ-AFF) 30%, Other specified Schizophrenia spectrum and other Psychotic disorders (OHER SCZ) 22%. Patients sample.** Gender: 60% male. LAI treatment: 68% of all patients treated with 2nd generation LAI.

**Table 2. GEE (Generalized Estimating Equations) models for within-subject correlations.**

| | hospitalization rates | total number of hospitalizations | emergency rates | total number of emergency visits | total number of planned outpatient visits |
|---|---|---|---|---|---|
| Treatment | $\chi^2 = 111.6$; p<0.001* | $\chi^2 = 156.4$; p<0.001* | $\chi^2 = 95.833$; p<0.001* | $\chi^2 = 50.213$; p<0.001* | $\chi^2 = 38.766$; p<0.001* |
| LAI generation | $\chi^2 = 8.014$; p = 0.005* | $\chi^2 = 3.737$; p = 0.053 | $\chi^2 = 6.435$; p = 0.011* | $\chi^2 = 8.966$; p = 0.003* | $\chi^2 = 1.254$; p = 0.263 |
| Age | $\chi^2 = 0.519$; p = 0.471 | $\chi^2 = 0.539$; p = 0.463 | $\chi^2 = 0.253$; p = 0.615 | $\chi^2 < 0.001$; p = 0.983 | $\chi^2 = 0.071$; p = 0.789 |
| Gender | $\chi^2 = 3.167$; p = 0.075 | $\chi^2 = 2.433$; p = 0.119 | $\chi^2 = 0.021$; p = 0.884 | $\chi^2 = 0.116$; p = 0.734 | $\chi^2 = 1.584$; p = 0.208 |
| educational level | $\chi^2 = 0.220$; p = 0.639 | $\chi^2 = 0.711$; p = 0.399 | $\chi^2 = 0.919$; p = 0.338 | $\chi^2 = 0.462$; p = 0.497 | $\chi^2 = 2.404$; p = 0.121 |
| Diagnosis | $\chi^2 = 2.477$; p = 0.290 | $\chi^2 = 1.879$; p = 0.391 | $\chi^2 = 1.725$; p = 0.422 | $\chi^2 = 0.530$; p = 0.767 | $\chi^2 = 3.354$; p = 0.187 |
| illness duration | $\chi^2 = 0.078$; p = 0.780 | $\chi^2 = 1.547$; p = 0.214 | $\chi^2 = 0.285$; p = 0.594 | $\chi^2 < 0.001$; p = 0.992 | $\chi^2 = 2.553$; p = 0.110 |

N = 207; Z = -9.769; p<0.001). Hospitalization rates (from 61.8% to 5.3%; McNemar Test; N = 207; χ2 = 113.076; p<0.001), emergency rates (from 66.7% to 24.2%; McNemar Test; N = 207; χ2 = 77.235; p<0.001), and emergency visits (from 337 to 106; Wilcoxon test; N = 205; Z = -9.109; p<0.001) also significantly decreased after the introduction of LAIs. On the contrary, planned outpatient visits significantly increased (from 6.4 to 9.1; Wilcoxon test; N = 205; Z = -6.125; p<0.001) (Fig 3).

The number of hospitalizations was significantly reduced by both first and second generation LAIs compared to oral antipsychotic treatment (Wilcoxon test; first generation LAI; N = 66; Z = -4.965; p<0.001; second-generation LAI; N = 141; Z = -8.427; p<0.001).

Hospitalization rates were significantly reduced by both first and second generation LAI compared to oral antipsychotic treatment (first-generation LAI; N = 66; $\chi^2 = 27.034$; p<0.001; second-generation LAI; N = 141; $\chi^2 = 84.100$; p<0.001).

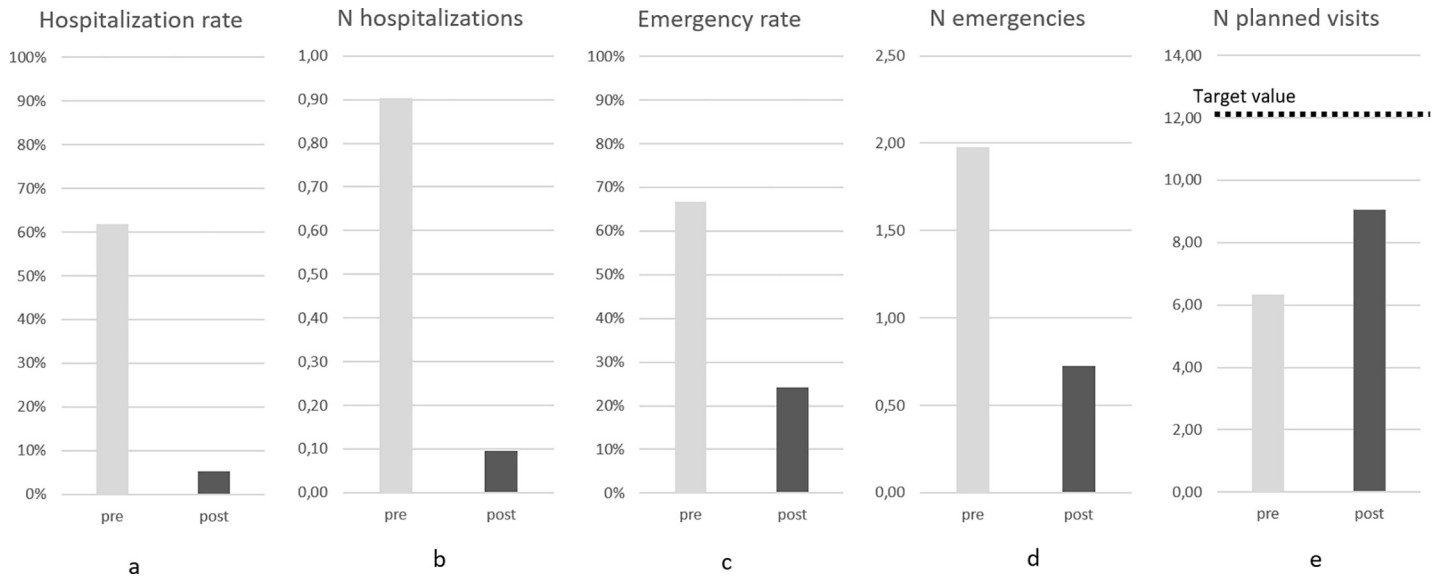

**Fig 3. Oral (pre) vs LAI (post) antipsychotic treatment effect on endpoints.** (a) Hospitalization rate: 61.8% pre vs. 5,3% post; (b) N of hospitalizations per patient per year: 90.3 pre vs. 9.8 post; (c) Emergency rate: 66.7% pre vs. 24.2% post; (d) N of emergency visits per patient per year: 1.63 pre vs. 0.52 post; (e) N of planned visits per patient per year (target value = 12): 6.4 pre vs. 9.1 post.

Again, the emergency visits were significantly reduced by both first and second generation LAIs compared to oral antipsychotic treatment (Wilcoxon test; first-generation LAI; N = 66; Z = -5.214; p<0.001; second-generation LAI; N = 139; Z = -7.565, p<0.001).

The emergency rates were significantly reduced by both first and second generation LAIs compared to oral antipsychotic treatment (first-generation LAI; N = 66; $\chi^2$ = 28.033; p<0.001; second-generation LAI; N = 141; $\chi^2$ = 47.779; p<0.001).

Planned outpatient visits significantly increased with both first and second generation LAIs compared to oral antipsychotic treatment (Wilcoxon test; first-generation LAI from 5.0 to 10.4 per patient per year; N = 66; Z = -5.191; p<0.001; second-generation LAI from 7.0 to 8.5 per patient per year; N = 139; Z = -3.563; p<0.001).

## First vs. second generation LAI antipsychotic treatment

Second generation LAIs were significantly more effective than first-generation LAIs on all primary endpoints, except emergency rates. The decrease of number of hospitalizations was significantly higher for second generation LAIs (first-generation LAI reduced from 50 to 4, second-generation LAI reduced from 137 to 16; Mann-Whitney U Test; N = 207; U = 3860; p = 0.033). The decrease of hospitalization rates is also significantly higher for second generation LAIs (first-generation LAI reduced from 47.0% to 3.0%, second-generation LAI reduced from 68.8% to 6.4%; Mann-Whitney U Test; N = 207; U = 3779; p = 0.011). The decrease of emergency visits is significantly higher for second generation LAIs (first-generation LAI reduced from 74 to 16, second-generation LAI reduced from 263 to 90; Mann-Whitney U Test; N = 205; U = 5347; p = 0.046) (Fig 4). The decrease of emergency rates is not significant (first-generation LAI reduced from 59.1% to 13.6%, second-generation LAI reduced from 70.2% to 29.1%; Mann-Whitney U Test; N = 207; U = 4527; p = 0.720). As to planned outpatient visits, after switching to LAIs, there is no statistically significant difference between the two generations of LAIs (Mann-Whitney U Test; N = 205; U = 3945; p = 0.105). However, since the number of planned outpatient visits was significantly lower for patients treated with first generation LAIs (Mann-Whitney U Test; N = 205; U = 3048.5; p<0.001), this leads to a higher statistically significant increase for first-generation LAIs (Mann-Whitney U Test; N = 205; U = 3115; p<0.001).

## Discussion

In our study, we considered the number of hospitalizations, the number of emergency and planned visits one year before and after the change from oral to LAI antipsychotics. For each patient, the effectiveness (i.e., the efficacy under ordinary circumstances and not under controlled circumstances) of LAIs was measured and compared [7]. Our findings, consistently with those reported in the literature, show that the efficacy of antipsychotic medications in enhanced by the use of LAI formulations. The main novelty of our study is the focus on the optimization of clinical processes and healthcare resources with LAIs, not only in terms of hospitalization rates but also of emergency and planned visits, which are rarely considered in studies on the effectiveness of antipsychotic medications.

Consistently with previous studies [10], all our primary endpoints show that the use of LAIs is significantly associated with a reduction of hospitalization rates and emergency visits. Moreover, the number of planned outpatient visits with physicians significantly increases after the introduction of LAIs, although we excluded the planned contacts with nurses for injection administration [11]. This finding clearly indicates that the use of LAIs is associated with a better distribution of resources, which should be taken in serious consideration by clinicians, policy makers and all stakeholders involved in the mental health field [12]. Contrary to what we

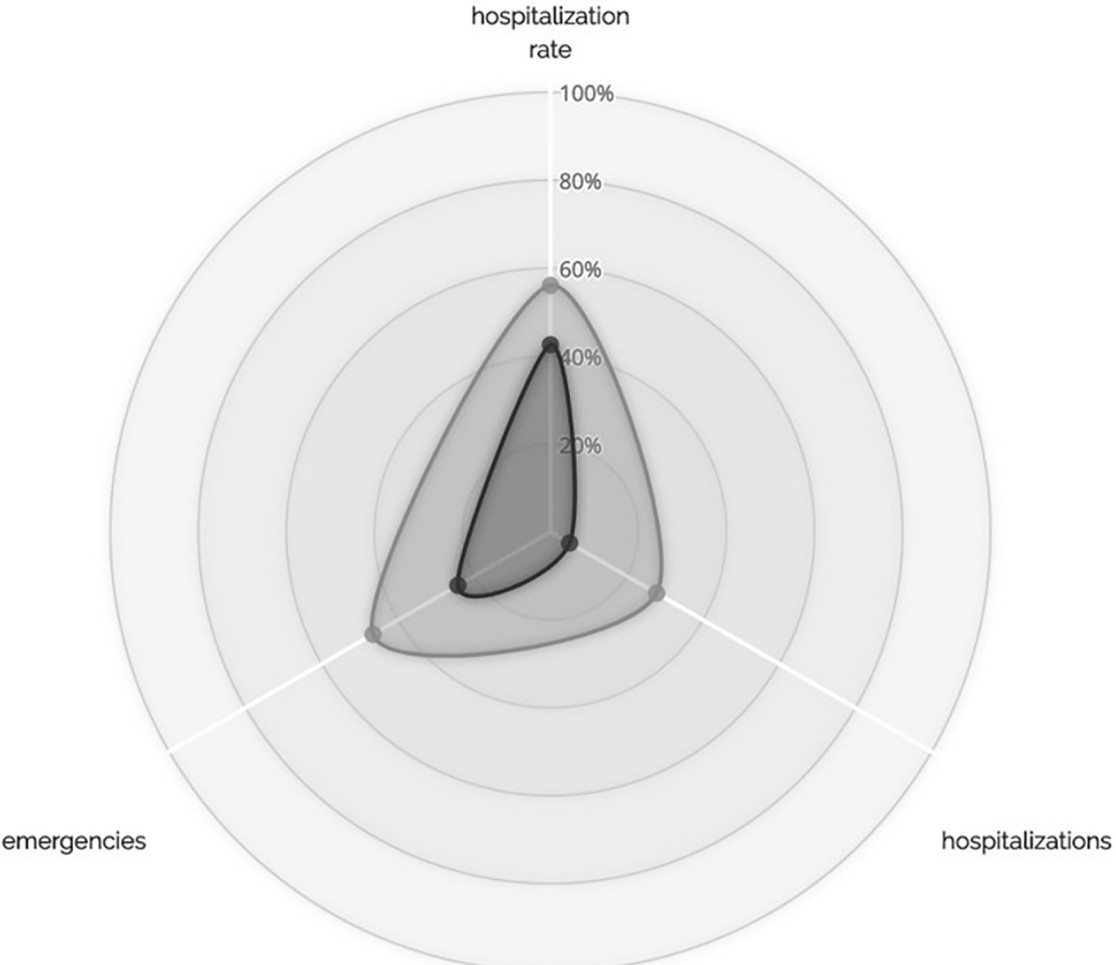

**Fig 4. Generation effect on primary endpoints.** 100% circle: oral treatment; light grey: first generation LAI, dark grey: second generation LAI. After T0: hospitalization rate is 56.1% first-generation LAI vs. 42.6% second-generation LAI of that with oral treatment; N of hospitalizations is 22.9% first-generation LAI vs. 5.0% second-generation LAI of that with oral treatment; emergencies are 46.5% first-generation LAI vs. 24.3% second-generation LAI of those with oral treatment.

could anticipate, the increase in the number of planned outpatient visits from 6.4 to 9.1 per patient per year suggests that the introduction of the monthly injection increases the number of contacts of patients with the local mental health center. This means that the use of LAI antipsychotics is associated with a more focus on patients' real-life needs and more time dedicated to psychosocial interventions, as suggested by most international guidelines [13,14]. We believe that, in case of infinite available resources, outpatient visits should be planned at least every six weeks for a better therapeutic alliance and for a better patient's motivation to join integrated treatments. The increase in planned visits correlates with better pharmacological adherence and rehospitalization prevention [15]. A better therapeutic alliance, due to increased planned visits, could be itself a more successful approach to relapse prevention [16,17]. As regards the possible effect of the hospitalization length of stay, the average duration of the hospitalization for schizophrenia spectrum disorder is 18.1 days in our Department of Mental Health. This value combined with the average number of hospitalizations per patient/year (0.903 for oral, 0.097 for LAI treatment) makes quite small the possibility that a long

period hospitalization may preclude a patient from readmission or emergency visits. More-over, these rare events would contribute to reducing the evidence of a significant reduction of readmission or emergency visits, which our results show."

Furthermore, we analyzed the effect of first versus second generation LAI antipsychotics on our endpoints. Several recent studies showed that second-generation LAI antipsychotics are superior to first-generation LAIs on treatment adherence (defined as the number of non-over-lapping days of supply divided by the number of days in the observational period of 365 days) [18] and rehospitalization risk [19]. To our knowledge, for the first time, our findings show that second-generation LAIs are more effective also in reducing the number of hospitalizations and emergency visits, although the same number of planned outpatient visits with first genera-tion LAIs. These data suggest that second-generation antipsychotic LAIs improve more effec-tively the clinical management of psychosis also when compared with first generation LAIs.

Of course, our study has some important limitations. A first important limitation is the choice of a mirror-image study design without a control group. The choice of a control group, i.e. patients with the same propensity score at T0 who continued on oral medication, is quite difficult because the true propensity score is never known in observational studies. Moreover, RCTs also present selection bias due to the enrolment of patients with different therapy adher-ence from real-world settings and, furthermore, in such design the trial itself could affect patient outcomes (Hawthorne effect), because of the social treatment and the increased per-sonal attention often associated with participating in trials [1]. Our design choice is supported by several other authors [1,18–21] who used studies designed without control groups. A sec-ond important limitation is that we took into account only one-way of switching for two rea-sons. We were not able to collect from medical records the number of patients discontinuing LAIs. Indeed this limitation in the collected data may bias the results positively. However, with a specific focus group, we estimated, between 10% and 15%, the number of participants who discontinued LAIs in our department. This data is compatible with what reported in the litera-ture with similar study designs [20, 21]. These selection biases represent limitations for nearly all pragmatic studies [11]. Nevertheless, even with their imperfections, these studies better reflect the broad range of patients in the "real-life" management of schizophrenia spectrum disorders which is fundamental for the assessment of clinical process management. A third limitation is that we did not collect any measure on patients' psychiatric symptoms and func-tionality. No subjective questionnaires on quality of life or satisfaction with therapy were given to patients, as seen in other studies. Given the naturalistic and retrospective design of this study, we only used illness duration as an indirect measure of illness severity. However, this study aims to evaluate clinical process management and not clinical outcomes. Moreover, bet-ter outcomes in clinical processes could indirectly suggest also better clinical response at least in terms of symptoms. Another limitation is the retrospective design of the study and the fact that the observation period was limited to one year, whereas a longer period could have pro-vided more information. This methodological choice was due to the fact that we wanted to analyze the effectiveness of LAI medications in the real world and one-year observation, albeit short for sophisticated trials, may be considered adequate for this type of studies.

Due to these limitations, we are aware that the study design is inadequate to draw causal conclusions about the effectiveness of LAI as opposed to oral antipsychotics, but an alternative interpretation of our results can be that LAIs are definitively more effective on the population of patients who need to switch to LAI in their clinical history. We can only suppose, because of the lack of data on the motivations for the switch, that the subjects of our study had shown bad therapeutic adherence, and therefore had been switched to LAI.

In conclusion, our study combines for the first time: (i) a retrospective, naturalistic and mirror design; (ii) data analysis from medical records using each patient as control of her/

himself, i.e. for each patient, the medical history before LAI is compared with her/his history after LAI [10]; and (iii) the analysis of the total number of emergency visits and the of planned outpatient visits. A key point is that data were collected from community health records, in order to control for confounding variables (e.g. chances in life context) that may influence the outcomes we considered in the analyses (i.e., treatment adherence, hospitalization rates, emergency visits).

These results suggest a relevant advantage by using LAIs for all stakeholders involved in the mental health field, including policy makers, mental health professionals, patients and caregivers. In particular, from the perspective of policy makers, the use of LAIs may result in marked savings given the significant reduction in the hospitalization rate [20]. Mental health professionals could optimize their work considering that emergency interventions require more resources in terms of time, employed staff and risk factors for professionals' and patients' safety (e.g., accidents). In fact, a more "virtuous" longitudinal observation of patients may reduce the risk of burnout in professionals. As regards the patients, a reduction in relapse rate improves prognosis and psychological personal burden, related to hospitalization treatments and may prevent family burden.

Finally, LAI antipsychotics actually reduce the severe economic burden of schizophrenia spectrum disorders, not only in terms of direct and indirect costs, but they can also improve other costs (e.g. the costs of justice and of law enforcement interventions), which are mainly related to emergency visits. We are currently evaluating the effect of LAI treatment on the economic costs of this mental disorder on society, and we will try to assess it in future work.

## Supporting information

**S1 Data.**
(ZIP)

## Author Contributions

**Conceptualization:** Valeria Latorre.

**Data curation:** Valeria Latorre, Michele Gattullo.

**Formal analysis:** Michele Gattullo, Antonio Emmanuele Uva.

**Investigation:** Valeria Latorre, Apostolos Papazacharias, Maria Lorusso, Gaetano Nappi, Paola Clemente, Antonia Spinelli, Giovanni Carrieri, Enrico D'Ambrosio.

**Methodology:** Valeria Latorre, Antonio Emmanuele Uva.

**Project administration:** Valeria Latorre, Domenico Semisa.

**Supervision:** Domenico Semisa.

**Validation:** Apostolos Papazacharias, Domenico Semisa.

**Visualization:** Michele Gattullo.

**Writing – original draft:** Valeria Latorre.

**Writing – review & editing:** Apostolos Papazacharias, Antonio Emmanuele Uva.

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
