## [Decision Letter · Decision Letter 0]

2 Dec 2019

PONE-D-19-20664

Improving the “real life” management of schizophrenia spectrum disorders by LAI antipsychotics: a one-year mirror-image retrospective study in community mental health services

PLOS ONE

Dear Dr latorre,

Thank you for submitting your manuscript to PLOS ONE. After careful consideration, we feel that it has merit but does not fully meet PLOS ONE’s publication criteria as it currently stands. Therefore, we invite you to submit a revised version of the manuscript that addresses the points raised during the review process.

We would appreciate receiving your revised manuscript by Jan 16 2020 11:59PM. To enhance the reproducibility of your results, we recommend that if applicable you deposit your laboratory protocols in protocols.io, where a protocol can be assigned its own identifier (DOI) such that it can be cited independently in the future. For instructions see: http://journals.plos.org/plosone/s/submission-guidelines#loc-laboratory-protocols

We look forward to receiving your revised manuscript.

Kind regards,

Kevin Lu, PhD

Academic Editor

PLOS ONE

Journal Requirements:

3. Thank you for providing the additional information regarding your ethics review at submission. In your methods section, please ensure that you discuss the ethical considerations. Based on the information you provided we would suggest the following: 'We informed the [INSERT SPECIFIC NAME OF ETHICS COMMITTEE] prior to initiating the study, in line with Istituto Superiore di Sanità protocol.  Each patient was assigned with an ID code to guarantee anonymity. All the patients whose data were collected had previously signed the informed consent, present in the medical record, to the processing of personal data and the use of the data for research purposes.

Reviewers' comments:

Reviewer's Responses to Questions

**Comments to the Author**

1. Is the manuscript technically sound, and do the data support the conclusions?

Reviewer #1: Yes

Reviewer #2: Partly

2. Has the statistical analysis been performed appropriately and rigorously? 

Reviewer #1: Yes

Reviewer #2: No

3. Have the authors made all data underlying the findings in their manuscript fully available?

Reviewer #1: Yes

Reviewer #2: Yes

4. Is the manuscript presented in an intelligible fashion and written in standard English?

Reviewer #1: Yes

Reviewer #2: Yes

5. Review Comments to the Author

Reviewer #1: The authors examined the effectiveness of LAIs compared to oral medications in patients with schizophrenia spectrum disorder. Their primary outcome measures were total number of hospitalizations, hospitalization rates, and total number of emergency visits. As a secondary outcome, the authors examined the number of planned outpatient visits as a measure of therapeutic compliance and alliance. These outcome measures were also examined between first and second-generation LAIs. The authors found that LAIs compared to oral medications significantly reduced hospitalizations (total number and rate) and increased the number of planned visits. They also report that second-generation LAIs are superior to first-generation LAIs in reducing the rates of hospitalization and emergency visits. Overall, the manuscript is well-written and concise. However, the following issues should be addressed:

1. The authors included 207 patients. Did the authors include all patients who switched from oral medication to LAIs or only patients who have 1-year follow-up data available? If the authors are only selecting cases with 1-year follow-up information available, the authors may be biasing the results by only selecting cases with good outcomes.

2. Related to the above comment, do the authors have information on the number of participants who switched to LAIs, but subsequently discontinued?

3. Introduction: Should it be 29,823 instead of 29.823?

4. Introduction: “These results are in contrast to meta-analysis of RCTs, which showed no superiority of LAIs” – the authors may want to clarify what they mean by superiority (e.g., in preventing relapse, hospitalizations, adherence, clinical outcomes?)

5. Methods, Study Sample: “Patients with inadequate data were excluded”. The authors may want to clarify what they mean by “inadequate”.

6. Methods: The authors considered hospitalization rates, total number of hospitalizations and total number of emergency visits as their main outcome measures. Did the authors also look at rates of emergency visits?

7. Methods/Results: In comparing first- and second-generation LAIs, I am wondering why the authors used Mann-Whitney U test for hospitalization rates instead of McNemar test.

8. I apologize if I missed this, but were the patients on any concomitant medications?

9. Results: The authors report total number of hospitalizations. Can the authors also provide the mean number of hospitalizations before and after T0?

10. Results: It will be helpful to include a table with the demographic and clinical characteristics of the included participants.

11. Results: Did the authors find any differences in demographic or clinical characteristics between patients who had first compared to second-generation LAIs?

Reviewer #2:

Dear Dr. Lu,

Thank you for the opportunity reviewing the manuscript “Improving the “real life” management of schizophrenia spectrum disorders by LAI antipsychotics: a one-year mirror-image retrospective study in community mental health services” submitted to the Plos One Journal.

In this paper, the authors examined the effectiveness of long-acting injectable antipsychotics (LAIs) compared to oral medications among patients with schizophrenia over a 10-year period. The authors found that LAIs, in particular, second generation ones, were associated with reduced hospitalization rates and emergency visits, at the same time improved the economic burden of schizophrenia. While the study design is relatively robust and findings important to clinical practices, there are a few limitations that need to be addressed before accepted for publication.

Major issues:

• The study design required subjects to be followed for a year before switching to LAIs and at least a year after T0. This could exclude patients who died or lost to follow-up over the post-T0 period. This immortal bias of a year could make the eligible subjects artificially better outcomes due to selection bias.

• Number of hospital admissions alone may not be sufficient in demonstrating the comparative effectiveness of LAIs and oral drugs. A person can be hospitalized for long periods, and technically preclude him from readmission/ emergency room visits. The total length of stay is an important outcome and should be interpreted along with other healthcare utilization patterns.

• Using non-parametric tests is insufficient to prove statistical differences in the treated group compared to themselves before switching to LAIs. One should consider the effects of other comorbidities which could affect hospitalization/emergency room visits. Also, due to within subject correlation, events from the same subjects should be considered in comparison. I would suggest using GEE models to account for within subject correlations.

Minor issues:

• The paper speaks of costs but without mentioning any measurement of costs in monetary terms. In order to be cost-effective, comparing number of hospitalization and ER visits are not enough. Per month per person costs and per ER visits costs should be analuzed, while accounting for differences in baseline medical needs.

• The terms of “hidden costs” is being used loosely and best avoid or specified.

6. PLOS authors have the option to publish the peer review history of their article (what does this mean?). If published, this will include your full peer review and any attached files.

Reviewer #1: No

Reviewer #2: No

---

## [Author Response · Author response to Decision Letter 0]

16 Jan 2020

REVIEWERS' COMMENTS:

Reviewer: 1

Overall, the manuscript is well-written and concise. However, the following issues should be addressed:

1. The authors included 207 patients. Did the authors include all patients who switched from oral medication to LAIs or only patients who have 1-year follow-up data available? If the authors are only selecting cases with 1-year follow-up information available, the authors may be biasing the results by only selecting cases with good outcomes.

2. Related to the above comment, do the authors have information on the number of participants who switched to LAIs, but subsequently discontinued?

Response

We thank the reviewer for this comment. As we already stated in the study limitations in the discussion section, we could not collect information on patients who discontinued LAI therapy. However, with a specific focus group, we estimated, between 10% and 15%, the number of participants who discontinued LAIs in our department, This data is compatible with what reported in the literature with similar study designs [20, 21]. We added and clarified in the revised version of the paper in the Discussion Section: 

“We were not able to collect from medical records the number of patients discontinuing LAIs. Indeed this limitation in the collected data may bias the results positively. However, with a specific focus group, we estimated, between 10% and 15%, the number of participants who discontinued LAIs in our department. This data is compatible with what reported in the literature with similar study designs [20, 21]. These selection biases represent limitations for nearly all pragmatic studies [11]. Nevertheless, even with their imperfections, these studies better reflect the broad range of patients in the “real-life” management of schizophrenia spectrum disorders which is fundamental for the assessment of clinical process management.”

3. Introduction: Should it be 29,823 instead of 29.823?

Response

We corrected the number format.

4. Introduction: “These results are in contrast to meta-analysis of RCTs, which showed no superiority of LAIs” – the authors may want to clarify what they mean by superiority (e.g., in preventing relapse, hospitalizations, adherence, clinical outcomes?)

Response

We clarified, in the revised version of the paper “…which showed no superiority of LAIs in preventing relapse and hospisalizations [8].” 

5. Methods, Study Sample: “Patients with inadequate data were excluded”. The authors may want to clarify what they mean by “inadequate”.

Response

We clarified, in the revised version of the paper: ” Patients with illegible medical records were excluded”

6. Methods: The authors considered hospitalization rates, total number of hospitalizations and total number of emergency visits as their main outcome measures. Did the authors also look at rates of emergency visits?

Response

We added it as a primary end-point as suggested by the reviewer. We report the text added in the revised paper and we also modified Figure 3 accordingly. Figure 4 was not modified because the difference in the variation of emergency rates between first- and second-generation LAI is not significant.

“We defined the primary end-points of the study: (i) hospitalization rates, (ii) total number of hospitalizations, (iii) emergency rates, and (iv) total number of emergency visits, as associated with the severity of the condition of the patient.”

“[…] emergency rates (from 66.7% to 24.2%; McNemar Test; N=207; χ2=77.235; p<0.001) also significantly decreased after the introduction of LAIs.”

“The emergency rates were significantly reduced by both first and second-generation LAIs compared to oral antipsychotic treatment (first-generation LAI; N=66; χ2=28.033; p<0.001; second-generation LAI; N=141; χ2=47.779; p<0.001).”

Comparing first- and second-generation LAIs “The decrease of emergency rates is not significant (first-generation LAI reduced from 59.1% to 13.6%, second-generation LAI reduced from 70.2% to 29.1%; Mann-Whitney U Test; N=207; U=4527; p=0.720).”

7. Methods/Results: In comparing first- and second-generation LAIs, I am wondering why the authors used Mann-Whitney U test for hospitalization rates instead of McNemar test.

Response

In these analyses, for each end-point, we considered as dependent variable the difference “d” between the value of the end-point before and after T0, and then we compared the two samples first and second-generation LAI. As to the hospitalization and emergency rate, we can have d=-1 if a patient was not hospedalized before T0 and hospedalized after T0, d=1 if a patient was hospedalized before T0 and not hospedalized after T0, d=0 in the other cases. Thus, we have no more two endpoints (hospitalization: yes or no) to analyze and then we cannot use McNemar test. 

8. I apologize if I missed this, but were the patients on any concomitant medications?

Response

We thank the reviewer for this comment. We explicitly excluded patients with LAI concomitant use of psychiatric oral therapy, but we did not report in the exclusion criteria. We had no information on other medical treatments.

We clarified, in the revised version of the paper in the exclusion criteria “b) LAI concomitant antipsychotic medication;”

9. Results: The authors report total number of hospitalizations. Can the authors also provide the mean number of hospitalizations before and after T0?

Response

We added, in the revised version of the paper: “the number of hospitalizations was drastically reduced from 187 (0.903 hospitalizations per patient/year) to 20 (0.097 hospitalizations per patient/year)”

10. Results: It will be helpful to include a table with the demographic and clinical characteristics of the included participants.

Response

We added the Table in the revised version of the paper “Table 1: Sociodemographic and clinical characteristics of our sample”

11. Results: Did the authors find any differences in demographic or clinical characteristics between patients who had first compared to second-generation LAIs? 

Response

We clarified, in the revised version of the paper that, using GEE, we found there is not enough evidence to conclude that age, gender, educational level, diagnosis, and illness duration have an effect on the end-points. See response to comment n.3 of reviewer 2.

 

Reviewer: 2

While the study design is relatively robust and findings important to clinical practices, there are a few limitations that need to be addressed before accepted for publication.

Major issues:

1 The study design required subjects to be followed for a year before switching to LAIs and at least a year after T0. This could exclude patients who died or lost to follow-up over the post-T0 period. This immortal bias of a year could make the eligible subjects artificially better outcomes due to selection bias.

Response

Please see response to Reviewer 1, point 1 and 2

2 Number of hospital admissions alone may not be sufficient in demonstrating the comparative effectiveness of LAIs and oral drugs. A person can be hospitalized for long periods, and technically preclude him from readmission/ emergency room visits. The total length of stay is an important outcome and should be interpreted along with other healthcare utilization patterns.

Response

We thank the reviewer for raising this issue. We agree that the total length of stay is an important factor. We discussed and clarified this effect, in the revised version of the paper in the discussion section:

“As regards the possible effect of the hospitalization length of stay, the average duration of the hospitalization for schizophrenia spectrum disorder is 18.1 days in our Department of Mental Health. This value combined with the average number of hospitalizations per patient/year (0.903 for oral, 0.097 for LAI treatment) makes quite small the possibility that a long period hospitalization may preclude a patient from readmission or emergency visits. Moreover, these rare events would contribute to reducing the evidence of a significant reduction of readmission or emergency visits, which our results show.”

3 Using non-parametric tests is insufficient to prove statistical differences in the treated group compared to themselves before switching to LAIs. One should consider the effects of other comorbidities which could affect hospitalization/emergency room visits. Also, due to within-subject correlation, events from the same subjects should be considered in comparison. I would suggest using GEE models to account for within-subject correlations.

Response

We thank the reviewer for having pointed out this issue. As suggested, we used GEE models to account for within-subject correlations considering the effects of all the variables on the end-points: treatment (before and after T0), LAI generation, age, gender, educational level, diagnosis, and illness duration. We added these results in Statistical Analyses:

“Effects of treatment (before and after T0), LAI generation, age, gender, educational level, diagnosis, and illness duration on the end-points were measured. We used GEE (Generalized Estimating Equations) models to account for within-subject correlations. These preliminary analyses revealed that the treatment and LAI generation seems to have major effects. Therefore, we studied in deep with specific tests the effects of treatment and LAI generation.”

and a new sub-section in Results:

“GEE models

GEE model analyses, whose results are reported in Table 2, revealed that the treatment has an effect on all the end-points. LAI generation seems to have an effect on hospitalization rates, emergency rates, and total number of emergency visits; it has a lower effect on total number of hospitalizations and no effect on total number of planned outpatient visits. As to the other characteristics, there is not enough evidence to conclude that they have an effect on the end-points.

Table 2: GEE (Generalized Estimating Equations) models for within-subject correlations.”

Minor issues:

4 The paper speaks of costs but without mentioning any measurement of costs in monetary terms. In order to be cost-effective, comparing number of hospitalization and ER visits are not enough. Per month per person costs and per ER visits costs should be analyzed, while accounting for differences in baseline medical needs.

Response

We are currently working on a specific paper on the cost assessment of LAI treatment. We added in the conclusions: “We are currently evaluating the effect of LAI treatment on economic costs of this mental disorder on society, and we will try to assess it in future work.” 

5 The terms of “hidden costs” is being used loosely and best avoid or specified.

Response

We thank the reviewer for this comment. We remove the term “hidden” and specify the type of costs in our context. “…they can also improve other costs (e.g. the costs of justice and of law enforcement interventions), which are mainly related to emergency visits.”

 

EDITOR’S COMMENTS:

We added in the Study Design:

“We informed the local ethical committee prior to initiating the study, in line with Istituto Superiore di Sanità protocol. Each patient was assigned with an ID code to guarantee anonymity. All the patients whose data were collected had previously signed the informed consent, present in the medical record, to the processing of personal data and the use of the data for research purposes. Given the naturalistic design of the study, the results remained purely observational and researchers did not influence the results in any way.”

---

## [Decision Letter · Decision Letter 1]

21 Feb 2020

Improving the “real life” management of schizophrenia spectrum disorders by LAI antipsychotics: a one-year mirror-image retrospective study in community mental health services

PONE-D-19-20664R1

Dear Dr. latorre,

We are pleased to inform you that your manuscript has been judged scientifically suitable for publication and will be formally accepted for publication once it complies with all outstanding technical requirements.

With kind regards,

Kevin Lu, PhD

Academic Editor

PLOS ONE

---

## [Editor Report · Acceptance letter]

26 Feb 2020

PONE-D-19-20664R1 

Improving the “real life” management of schizophrenia spectrum disorders by LAI antipsychotics: a one-year mirror-image retrospective study in community mental health services 

Dear Dr. latorre:

I am pleased to inform you that your manuscript has been deemed suitable for publication in PLOS ONE. Congratulations! Your manuscript is now with our production department. 

With kind regards,

on behalf of

Professor Kevin Lu 

Academic Editor

PLOS ONE